# κ-Selenocarrageenan Oligosaccharides Prepared by Deep-Sea Enzyme Alleviate Inflammatory Responses and Modulate Gut Microbiota in Ulcerative Colitis Mice

**DOI:** 10.3390/ijms24054672

**Published:** 2023-02-28

**Authors:** Kai Wang, Ling Qin, Junhan Cao, Liping Zhang, Ming Liu, Changfeng Qu, Jinlai Miao

**Affiliations:** 1Key Laboratory of Marine Drugs, Ministry of Education, School of Medicine and Pharmacy, Ocean University of China, Qingdao 266003, China; 2Key Laboratory of Marine Eco-Environmental Science and Technology, First Institute of Oceanography, Ministry of Natural Resources, Qingdao 266061, China; 3Laboratory for Marine Drugs and Bioproducts, Qingdao Pilot National Laboratory for Marine Science and Technology, Qingdao 266237, China; 4Marine Natural Products R&D Laboratory, Qingdao Key Laboratory, Qingdao 266061, China

**Keywords:** heterologous expression, enzymatic preparation, κ-selenocarrageenan oligosaccharides, structural characterization, ulcerative colitis, gut microbiota

## Abstract

κ-Selenocarrageenan (KSC) is an organic selenium (Se) polysaccharide. There has been no report of an enzyme that can degrade κ-selenocarrageenan to κ-selenocarrageenan oligosaccharides (KSCOs). This study explored an enzyme, κ-selenocarrageenase (SeCar), from deep-sea bacteria and produced heterologously in *Escherichia coli*, which degraded KSC to KSCOs. Chemical and spectroscopic analyses demonstrated that purified KSCOs in hydrolysates were composed mainly of selenium-galactobiose. Organic selenium foods through dietary supplementation could help regulate inflammatory bowel diseases (IBD). This study discussed the effects of KSCOs on dextran sulfate sodium (DSS)-induced ulcerative colitis (UC) in C57BL/6 mice. The results showed that KSCOs alleviated the symptoms of UC and suppressed colonic inflammation by reducing the activity of myeloperoxidase (MPO) and regulating the unbalanced secretion of inflammatory cytokines (tumor necrosis factor (TNF)-α, interleukin (IL)-6 and IL-10). Furthermore, KSCOs treatment regulated the composition of gut microbiota, enriched the genera *Bifidobacterium*, *Lachnospiraceae_NK4A136_group* and *Ruminococcus* and inhibited *Dubosiella*, *Turicibacter* and *Romboutsia*. These findings proved that KSCOs obtained by enzymatic degradation could be utilized to prevent or treat UC.

## 1. Introduction

Carrageenan is a sulfated linear polysaccharide extracted from the cell wall of red algae. Based on the difference in the number of sulfate groups and the presence of 3,6-anhydro-α-D-galactopyranosyl (3,6-AG), carrageenans are further classified into κ-, ι- and λ-carrageenans [1]. κ-Carrageenan is alternately composed of 4-linked-α-D-3,6-anhydrogalactose (DA) and 3-linked-4-O-sulfated-β-D-galactopyranose (G4S), which has been recognized as safe by the U.S. Food and Drug Administration [2,3]. However, its application is limited due to poor solubility and low bioavailability [4]. κ-Carrageenan oligosaccharides obtained by κ-carrageenan degradation can greatly improve these properties. Moreover, κ-carrageenan oligosaccharides exhibited antioxidant [5], anticoagulation [6] and antitumor effects [7].

It is well known that Se is an indispensable trace element for human health and can only be obtained from food. KSC is a kind of Se polysaccharide made from natural κ-carrageenan, in which Se partially replaces sulfur (S) [8]. It is reported that KSC had an immunomodulatory function and inhibited tumor growth in H22 tumor-bearing mice [9]. Theoretically, low molecular weight KSCOs hydrolyzed by KSC possess remarkable bioactivity. At present, KSCOs were created chemically using sodium selenite and κ-carrageenan oligosaccharides, but the unstable structure of products makes this process unsuitable for large production. In contrast, enzymatic hydrolysis yields products with a controlled structure and no contamination, which is now the preferred method for oligosaccharides production. However, the enzyme hydrolyzing KSC to KSCOs has rarely been researched. In the previous study, we described a potential κ-selenocarrageenase isolated from the cold seep in the South China Sea [8]. Here, this κ-selenocarrageenase was expressed in *Escherichia coli* and its degradation activity was demonstrated. Therefore, a novel and easy strategy for the utilization of KSC to produce functional KSCOs was provided.

UC is a chronic and recurrent inflammation of the intestine with a high incidence in Western countries [10,11]. The pathogenesis of UC is thought to be related to genetic susceptibility, immunity, environment and intestinal mucosal barrier loss [12]. The main clinical symptoms of UC include abdominal pain, diarrhea, bloody mucus and purulent stools [13,14]. It is worth noting that UC increases the risk of colorectal cancer, the third most common malignant tumor in the world [15]. Nevertheless, current drugs used to treat UC, such as aminosalicylate and mesalazine, tend to decline in response to treatment over time and lead to disease complications [11]. In addition, such drugs may induce adverse reactions, such as dilated cardiomyopathy and severe heart failure [16]. Therefore, it is urgent to develop new therapeutic drugs. In fact, nutrition plays a crucial role in preventing IBD [17]. Nutritional deficiencies, including micronutrients, are common in patients with IBD [18,19]. It has been demonstrated that dietary Se supplementation enhanced intestinal antioxidant function and relieved inflammation [20]. On the other hand, previous studies have shown that carrageenan oligosaccharides had potent effects on inhibiting the release of inflammatory cytokines [21,22,23]. However, the beneficial effects of KSCOs remain unclear for IBDs, such as UC.

In this work, we heterologously expressed and characterized a κ-selenocarrageenase from a marine bacterium named *Bacillus* sp. N1-1. The structure of KSCOs obtained from κ-selenocarrageenase hydrolysis of KSC was analyzed. KSCOs possess the activity of both selenium and κ-carrageenan oligosaccharides. Thus, we speculated that KSCOs may have effects on the treatment of UC. DSS is a polymer of anhydroglucose that induces UC when introduced through drinking water in rodents, such as guinea pigs, rabbits and mice [24,25]. This chemical compound is now widely used in basic research related to colitis. In this study, we aimed to explore the effects of KSCOs on DSS-induced UC in mice and investigated the underlying mechanism of action. 

## 2. Results

### 2.1. Enzymology Experiment

#### 2.1.1. Bioinformatics Analysis of SeCar

As our previous study mentioned, a deep-sea bacterium *Bacillus* sp. N1-1 has been preliminarily demonstrated to degrade κ-selenocarrageenan [8]. The SeCar gene (GenBank accession number: MW366920) from N1-1 genome was predicted as a candidate κ-selenocarrageenase as it was noted to be coding a putative glycoside hydrolase 16 (GH 16) protein. The open reading frame (ORF) of this gene consisted of 2184 bp and encoded 728 deduced amino acid residues, the first 25 amino acid residues of which were identified as a signal peptide sequence. The theoretical molecular weight of the mature protein was 79.51 kDa and the predicted isoelectric point was 4.40. It was predicted to be a stable hydrophilic protein with mean hydrophilicity (gravy) of −0.735, fat coefficient of 66.46 and instability index of 33.46. According to the conserved domain analysis (https://www.ncbi.nlm.nih.gov/Structure/cdd/wrpsb.cgi, accessed on 16 August 2020), the complete sequence of SeCar is mainly composed of four domains, of which the amino acid residues Arg153-Lys353 belongs to the GH16 family domain. GH16 family is concluded as a polyspecific glycoside hydrolase family and contains different enzymes, including κ-carrageenase, β-agarase, β-porphyranase, licheninase and laminarinase [26,27]. Multiple sequence alignment was carried out between SeCar and other reported GH 16 family κ-carrageenases (Appendix A). On the basis of alignment results, SeCar contained the conserved region ExDxxE, which is responsible for the double displacement mechanism in κ-carrageenase catalysis [28,29]. The above bioinformatics analysis elucidated the characteristics of SeCar as a κ-carrageenase. Additionally, the BLASTP analysis showed that SeCar shared the highest sequence identity of 28.12% with the κ-carrageenase of *Pseudoalteromonas tetraodonis* JAM-K142 among all characterized proteins [30].

#### 2.1.2. Expression and Purification of SeCar

For better characterization, the SeCar gene was cloned and expressed successfully in Escherichia coli. It was shown that the purified κ-selenocarrageenase was analyzed by SDS-PAGE in Appendix A. After the gene fused with (His)6-tag was expressed, the molecular weight of the purified recombinant protein was approximately 80 kDa, which was larger than the theoretical molecular weight (79.51 kDa). The activity of purified recombinant SeCar was 133 U/mg, which was much higher than that of the wild enzyme (18.58 U/mg).

#### 2.1.3. Biochemical Properties of SeCar

Figure 1A shows that the optimal temperature of purified SeCar was 40 °C. In addition, its activity remained stable at 20 °C, and 80% of its initial activity was maintained at 30 °C for up to 2 h (Figure 1B). The thermal stability of SeCar facilitates its storage and biotransformation in industrial production. The effects of various metal ions and chemical reagents on SeCar activity are shown in Figure 1C. K^+^ and Mn^2+^ slightly stimulated the enzyme activity. Cu^2+^, Fe^2+^ and Fe^3+^ inhibited the enzyme activity, among which Cu^2+^ had the greatest inhibitory effect, causing 80% of the enzyme activity impaired. The kinetic parameters of purified SeCar were determined using κ-selenocarrageenan as the substrate. The V_max_ and K_m_ values were 12.0048 mg/(mL·min) and 0.2389 mg/mL, respectively (Figure 1D), indicating that the κ-selenocarrageenase SeCar showed high affinity to the κ-selenocarrageenan.

### 2.2. Determination and Evaluation of KSCOs Structure

According to the high-performance gel permeation chromatography (HPGPC) spectra (Appendix A) and the detailed values (Appendix A), the KSCOs were mainly distributed below 1500 Da, among which 37.14% were 1379.49 Da and 31.69% were 816.82 Da.

#### 2.2.1. Electrospray Mass Spectrometry (ESI-MS) Analysis

To clarify the structure of KSCOs, MS analysis at negative ESI mode was conducted. The MS image of KSCOs (Appendix A) revealed peaks at *m*/*z* 437 and *m*/*z* 546, corresponding to [(DA-G4Se)]^−^ and [(DASe-G4Se)]^−^, respectively. The over selenated disaccharide units of DASe-G4Se are attributed to the mixing of ι-carrageenan in commercial κ-carrageenan [31,32]. The disaccharide unit in ι-carrageenan contains two sulfate groups, which might be replaced by selenate. Combined with the result of thin layer chromatography (TLC) analysis (Appendix A), we speculated that the peaks at *m*/*z* 341.1, *m*/*z* 665.2 and *m*/*z* 989 were representative of (DA-G4)^−^, [(DA-G4)_2_]^−^ and [(DA-G4)_3_]^−^, respectively, without carrying the selenate group. This deselenylation was possibly caused by the high cone voltage in the mass spectrometer [31,33].

#### 2.2.2. Fourier Transform Infrared (FTIR) Spectroscopy Analysis

The FTIR spectra analysis of KSCOs was shown in Figure 2. The intense peak at 3283 cm^−1^ was ascribed to the stretching vibration of O-H. The weak stretching band near 2925 cm^−1^ was ascribed to the stretching vibration of C-H. The peak at approximately 1598 cm^−1^ was associated with the stretching vibration of C=O. In addition to characteristic absorption peaks of polysaccharides, the peak near 1250 cm^−1^ was ascribed to the stretching vibration of S=O, indicating that the sulfate groups in κ-selenocarrageenan were not completely replaced. However, due to the selenylation modification, the absorption peaks near 1375 cm^−1^ and 762 cm^−1^ were attributed to the Se=O asymmetric stretching and C–O–Se symmetric vibrations, respectively [34]. Additionally, a strong absorption near 1024 cm^−1^ was assigned to the stretching vibration of the C–O–C glycosidic bond, indicating a pyranose unit in the carrageenan basic structure [35].

#### 2.2.3. Nuclear Magnetic Resonance (NMR) Spectroscopy

In the ^1^H NMR spectrum of the KSCOs (Figure 3A), there were signals of α and β configurations at the reducing end of G4S. The signal at δ 5.39 ppm was attributed to G4S-H-1α, while the chemical shift signal of G4S-H-1β appeared at δ 4.67 ppm [36]. Since selenylation occurred at C-4, the chemical shift of H-4 after selenite moved to the low field near δ 4.86 ppm. However, due to the overlap with the hydrogen signal in the solvent HOD, the chemical shift was not obvious. It has been reported that the signal δ 5.25 ppm was attributed to the H-1 of DA [37]. In this study, DA-H-3 and DA-H-5 were located in the region of δ 4.67 ppm and δ 4.17 ppm, respectively, due to the dehydration reaction at C-3 and C-6 of DA. As shown in Figure 3B, there were four anomeric carbon signals, which were 101.6, 101.3, 97.4 and 93.5 ppm, respectively. κ-Carrageenan is an alternating galactan of 1,3-linked β-D-galactopyranose 4-O-sulfate and 1,4-linked 3,6 anhydro-α-D-galactopyranose [2]. The anomeric carbon of β-D galactose was more than 100.0 ppm, while the terminal carbon of α-D galactose was less than 100.0 ppm. Therefore, the signals at 101.6 and 101.3 ppm were attributed to →3)-β-G4s-(1→ and →3)-β-G4Se-(1→ anomeric carbon. At 97.4 ppm, it was →4)-α-DA (1→ anomeric carbon signal; 93.5 ppm was →3)-G4Srα reducing anomeric carbon signal. The high field 62.1 ppm was →3)-β -G4s -(1→ C-6 signal. All chemical shifts were summarized in Table 1 and Table 2.

The ^1^H and ^13^C spectra of KSCOs were analyzed, and it was found that selenylation had no significant influence on the basic structure of κ-carrageenan, which was consistent with the previous report [38]. Since no substitution of the C-6 position was found in the DEPT 135° spectrum, we speculated that selenylation did not occur in position C-6 of →4)-α-DA-(1→. Therefore, combining ESI-MS, FTIR and NMR data, the selenium oligosaccharides in KSCOs were mainly composed of selenium-galactobioses and the predicted structure was shown in Appendix A.

### 2.3. Effects of KSCOs on the UC Mice

#### 2.3.1. KSCOs Relieved Symptoms of UC

The degree of UC in mice was assessed through body weight, disease activity index (DAI) and colon length. There was a significant decrease in the body weight of the DSS mice in this study (*p* < 0.001) (Figure 4A). KSCOs exhibited significant improvement in body weight loss (*p* < 0.001). Additionally, as shown in Figure 4B, mice treated with KSCOs exhibited an improved health status compared to mice with only DSS according to DAI. Furthermore, compared with DSS only, KSCOs treatment reduced the shortening of the colon significantly in mice (*p* < 0.01) (Figure 4C,D). According to the morphological examination (Figure 5A), colon tissues of the DSS group showed obvious erosion, goblet cell disappearance and inflammatory cell infiltration compared with the intact inner wall of the normal group, while KSCOs treatment alleviated these pathological changes of colonic tissue in colitis. The above phenomenon revealed that KSCOs relieved the systemic (weight loss and DAI) and local (CL shortened and HDS) symptoms of UC.

#### 2.3.2. KSCOs Regulated the Inflammatory Responses

As shown in Figure 5B, MPO activity was significantly activated in the colon tissue of the DSS group (*p* < 0.001), indicating an excessive inflammatory response. However, KSCOs reduced MPO activity dramatically (*p* < 0.001) compared with the DSS group. In addition, we measured the contents of proinflammatory cytokines including TNF-α and IL-6 and the anti-inflammatory cytokine IL-10 in serum. As shown in Figure 5C,D, compared to the normal group, DSS exposure increased the contents of TNF-α (*p* < 0.01) and IL-6 (*p* < 0.001) significantly, while it reduced the content of IL-10 (*p* < 0.001).

#### 2.3.3. KSCOs Reshaped the Composition of Gut Microbiota

A total of 713,927 sequences were obtained from 18 samples among the normal group, DSS group, and KSCOs group. Richness (Ace index) and diversity (Shannon and Simpson indices) of microbial communities were shown by alpha-diversity analysis (Appendix A). The Ace, Shannon, and Simpson indices in the DSS group all displayed a decline when compared to the normal group. Although the Ace, Shannon, and Simpson indices did not significantly increase following the administration of KSCOs in comparison to the DSS group, the increase in gut microbial richness and diversity was partially explained. The rarefaction curves tended to be saturated platforms (Appendix A), which indicated that the majority of the microbial diversity had been collected and the sequencing coverage was adequate.

As shown in Figure 6A, gut microbiota of mice in the three groups were mainly composed of Firmicutes and Bacteroidota at the phylum level. However, administration of KSCOs decreased the relative abundance of Firmicutes while increasing the relative abundance of Bacteroidota in DSS-induced colitis mice. In general, compared with the normal group, DSS significantly increased the ratio of Firmicutes to Bacteroidota (F/B) (*p* < 0.05), while this phenomenon was significantly reversed by KSCOs (*p* < 0.05) (Figure 6C). To further assess the predominant bacterial communities in the intestine across the three groups, linear discriminant analysis (LDA) and effect size (LefSe) was carried out. The generated cladogram reflected different gut microbiota compositions among mice from all groups (Figure 7A). The LDA discriminant histogram counted the microbial taxa with significant effects in multiple groups. Greater relative species abundance is represented by higher LDA scores. Via LDA scores, the findings revealed that *Bifidobacterium*, *Lachnospirace-ae NK4A136 group*, and *Ruminococcus* were prevalent in the KSCOs group while *Dubosiella*, *Turicibacter* and *Romboutsia* were prominent in the DSS group (Figure 7B). Specific differences between groups were evaluated at the genus level to further illustrate how KSCOs treatment affected the composition of gut microbiota (Figure 6B). At the genus level, compared to DSS group, KSCOs administration significantly enhanced the relative abundance of *Bifidobacterium*, *Lachnospiraceae_NK4A136_group* and *Ruminococcus* (Figure 7D–F). Additionally, compared to the normal group, the relative abundance of *Dubosiella* (*p* < 0.001), *Turicibacter* (*p* < 0.01) and *Romboutsia* (*p* < 0.01) increased significantly in the DSS group, while this increase was inhibited by KSCOs administration (Figure 7G–I). Acetate, propionate, butyrate and total SCFA concentrations were all considerably lower after receiving DSS without treatment, as shown in Figure 8 (*p* < 0.001, *p* < 0.05, *p* < 0.01 and *p* < 0.001, respectively). However, compared with the DSS group, KSCOs increased the concentration of butyrate significantly (*p* < 0.05) and tended to promote the biosynthesis of acetate and propionate.

## 3. Discussion

KSC is a marine selenium polysaccharide synthesized by selenization modification of κ-carrageenan, which has been included in the national safety standard for the use of the food nutrition fortification standard [39]. However, KSC has a high molecular weight and low bioavailability. The chemical or physical degradation process of KSC is uncontrollable, and the structure of degradation products is unstable. To date, there have been few studies on the hydrolysis of KSC by κ-selenocarrageenanase. In this study, we prepared KSCOs from a κ-selenocarrageenanase named SeCar. The novelty of the SeCar sequence suggests that it may exhibit properties distinct from other κ-carrageenases. It is worth noting that this is the first demonstration of KSC degradation by a κ-carrageenase.

There are multiple factors contributing to the pathogenesis of IBD, including the influence of micronutrients [40]. Summarizing recent reviews, Se exhibited an important role in the pathogenesis of IBD and Se deficiency was common in IBD patients [20,41]. Hence, the essential trace element Se has been drawn more attention for IBD prevention and treatment. Compared with inorganic Se, organic Se possesses lower toxicity and higher bioavailability. Here, we investigated the effects of KSCOs on DSS-induced colitis. According to the acceptable upper limit of adult Se intake (400 μg/d) recommended by WHO (2004) and Chinese Nutrition Society (2013), the doses of KSCOs were designed as 1.6, 3.2 and 6.4 mg/kg, which were equivalent to 25.5, 51 and 102 μg/kg of oral Se in mice, respectively [42]. The results showed that KSCOs relieved the systemic (weight loss and DAI) and local (CL shortened and HDS) symptoms of UC. MPO is a proinflammatory oxidase secreted by neutrophils and macrophages, which can destroy intestinal mucosal cells and cause inflammatory responses; therefore, it usually shows high activity in UC patients [43]. Additionally, after the occurrence of colitis, proinflammatory cytokines, such as TNF-α, IL-6 and IL-1β, are secreted and accumulated in large quantities due to the excessive activation of immune cells. These cytokines directly caused mucosal and tissue damage, triggering disease-specific inflammatory responses in colitis [44]. Regulating the secretion of these cytokines is extremely important for alleviating the inflammatory responses in colitis. Therefore, KSCOs could reduce inflammatory responses in UC mice via ameliorating neutrophil infiltration and regulating the level of inflammatory cytokines (TNF-α, IL-6 and IL-10).

The gut microbiota is considered as an important factor influencing the occurrence and severity of DSS induced colitis [45]. The positive effects of dietary Se supplementation on intestinal inflammation have been well demonstrated [40,46]. Moreover, as previously reported, at least part of the mechanism was due to Se altering the gut microbiota rather than directly affecting the gut [47]. To identify whether KSCOs regulates gut microbiota, 16S rRNA sequencing in fecal bacteria DNA was conducted and the high dose (6.4 mg/kg) group of KSCOs was selected to be sequenced. In this research, it can be found that dietary selenium KSCOs regulated the diversity and composition of gut microbiota in the DSS-induced mice, consistent with previous reports [48,49]. Specifically, KSCOs showed a function of reducing the ratio of Firmicutes to Bacteroidota (F/B). F/B is commonly denoted as the degree of dysbiosis in IBD [50,51], and a high proportion of Bacteroidota is associated with the resistance to inflammation [52]. Thus, it can be indicated that KSCOs could restore intestinal homeostasis by regulating the abundance of Firmicutes and Bacteroidota.

At the genus level, KSCOs enhanced the abundance of *Bifidobacterium*, *Lachnospiraceae_NK4A136_group* and *Ruminococcus*. *Bifidobacterium* is recognized as a probiotic, promoting intestinal health in the following aspects. In the intestine, *Bifidobacterium* can synthesize exopolysaccharides as the fermentation substrate of microbiota, which is beneficial to intestinal health [53,54]. Additionally, *Bifidobacterium* can enhance intestinal epithelial barrier function through metabolites and inhibit the inflammatory responses [55,56]. Furthermore, *Bifidobacterium*, *Lachnospiraceae_NK4A136_group* and *Ruminococcus* were reported to promote the production of SCFAs, which were capable of maintaining epithelial health and immune balance of the intestine [57,58,59]. KSCOs administration inhibited the growth of harmful bacteria, such as *Dubosiella*, *Turicibacter* and *Romboutsia*. The trends in the relative abundance of *Dubosiella* between groups were consistent with previous reports about UC [60,61,62]. However, more research is needed to determine the effect of *Dubosiella* on colitis. Increases in both *Turicibacter* and *Romboutsia* are associated with the development of colitis. It has been reported that *Turicibacter* with high abundance aggravated intestinal damage and led to serious complications [63]. Moreover, *Romboutsia* is considered as a pathogen, and its abundance is increased in many diseases, such as neurodevelopmental disorders [64], irritable bowel syndrome [65] and gastric cancer [66]. It can be found that the abundance of *Romboutsia* was increased in the intestine of DSS-induced colitis mice compared with that of healthy mice in this study, consistent with views in relevant studies [67,68,69]. SCFA has also been reported to ameliorate colitis through suppressing proinflammatory cytokines, such as TNF-α and IL-6 [70,71]. The reason for the higher SCFA content in the KSCOs group compared with the DSS group might be due to enriched *Bifidobacterium*, *Lachnospiraceae_NK4A136_group* and *Ruminococcus*. Moreover, a previous report found that organic sources of Se promoted the biosynthesis of propionate and butyrate [72]. There have been many studies on the remodeling of gut microbiota by different Se sources, such as selenium-enriched yeast [72], selenium-enriched probiotics [49] and selenium-containing tea polysaccharides [73]. The mechanism, however, is complex and few studies have clarified it. In this study, we described the effects of KSCOs on the gut microbiota in mice for the first time, but the role of Se in gut microbiota needs to be further explored in subsequent research. Taken together, KSCOs might alter the composition and metabolites of gut microbiota to relieve DSS-induced colitis.

## 4. Materials and Methods

### 4.1. Enzymology Experiment

The κ-selenocarrageenase (SeCar) gene (Locus_tag: N1.1_GM000361) was obtained from the whole genome of *Bacillus* sp. N1-1 (GenBank accession number: CP046564). κ-Selenocarrageenan was purchased from Qingdao Pengyang Biological Engineering Co., Ltd., Qingdao, China.

#### 4.1.1. Bioinformatics Analysis

Bioinformatics prediction and analysis of the amino acid sequence were carried out online. Physicochemical properties of amino acids were predicted using ExPASyProtparam (https://web.expasy.org/protparam/, accessed on 16 August 2020). The hydrophobicity of protein was predicted by ExPASyScale (https://web.expasy.org/protscale/, accessed on 16 August 2020). The prediction of signal peptide sequence was used by SignalP 5.0 Server (http://www.cbs.dtu.dk/services/SignalP/, accessed on 16 August 2020). Alignments of the amino acid sequences and other κ-carrageenases in NCBI database were performed using ClustalX (Version 1.8).

#### 4.1.2. Expression and Purification

Genomic DNA of *Bacillus* sp. N1-1 was extracted using the FastPure Bacteria DNA Isolation Mini Kit (Vazyme Biotech, Nanjing, China). The gene SeCar without the predicted signal sequence was amplified by PCR using the forward and reverse primers 5′-CACGAAAAAGAAAAAGATAATAATAAAAGTGAAC-3′ and 5′-CGTTACGCCTTCAATCGTAAC-3′. SeCar was cloned and ligated into pEASY-blunt E2 vector (TransGen Biotech, Beijing, China) to conduct recombinant plasmid. The constructed plasmid was transformed into BL21(DE3) competent cells (TransGen Biotech, Beijing, China) and then screened on Luria-Bertani (LB) medium supplemented with ampicillin. After incubation for 10 h, the positive colony was selected and cultured in LB medium with ampicillin in a shaker at 180 rpm at 37 °C until the absorbance value of bacterial solution reached OD_600_ = 0.8. The enzyme was prepared by adding isopropyl-beta-D-thiogalactopyranoside into recombinant *Escherichia coli* culture and then shaken at 150 rpm for 12 h at 16 °C. Cells were pelleted (7500× *g*; 15 min), resuspended in 50 mL of phosphate buffered saline (PBS), and lysed on ice by sonicating (300 w, 20 min). The supernatant after centrifugation was the crude enzyme of SeCar and was purified by Ni-affinity chromatography. The methods of gene expression and protein purification refer to the previous description [74,75].

#### 4.1.3. Biochemical Properties

Coomassie brilliant blue binding method was used to determine the total protein concentration. The enzyme activity was determined by 3,5-dinitrosalicylic acid (DNS) method with galactose as standard [76]. The amount of enzyme releasing 1 µmol galactose per minute under standard conditions is defined as one unit (U) of enzyme activity.

The optimum reaction temperature was determined by measuring the activity of SeCar in the range of 20 °C to 80 °C with 0.1% κ-selenocarrageenan as substrate. SeCar was incubated in PBS buffer at 20–60 °C for 0–24 h, and the residual activity was detected to assess thermostability. The optimal pH for SeCar activity was determined using different buffers, such as sodium citrate buffer (pH 3.0–6.0), phosphate buffer (pH 6.0–8.0), Tris-HCl buffer (pH 8.0–9.0) and glycine buffer (pH 9.0–11.0), at 40 °C with 0.1% (*w*/*v*) κ-selenocarrageenan as the substrate. SeCar was preincubated with the above buffers at 20 °C for 2 h, and the residual enzyme activity was detected to assess pH stability.

In order to evaluate the effects of metal ions and chemical reagents on SeCar, the enzyme assay was performed in the presence of 5 mM Na^+^, K^+^, Cu^2+^, Mg^2+^, Mn^2+^, Ca^2+^, Fe^2+^, Fe^3+^ and EDTA. Enzyme activity was measured at 40 °C and pH 7.0. The reaction without adding metal ions and chemical reagents was used as the control.

For the values of K_m_ and V_max_, the purified enzyme reacted with 0.025–0.2% κ-selenocarrageenan as substrate at 40 °C for 30 min, which were calculated by double reciprocal plotting. All of the above activity assays were performed in 3 replicates.

### 4.2. Isolation of the KSCOs

The KSCOs were prepared and isolated according to the previously described method with modifications [77,78]. The reaction system, containing 6 U of purified SeCar and 25 mM KSC, was conducted at 40 °C for 12 h. The lysate was boiled for 10 min to inactivate κ-selenocarrageenase and centrifuged for 15 min at 6000 r/min to remove impurities. Finally, four volumes of 95% ethanol (*v*/*v*) were added to precipitate the undegraded KSC. After centrifugation at 10,000 r/min, the supernatant was concentrated using a rotary evaporator at 60 °C and then lyophilized under vacuum at −60 °C to obtain crude KSCOs.

### 4.3. Molecular Weight of KSCOs

The molecular weight (MW) of KSCOs was evaluated by HPGPC [79]. The analysis was performed on a high-performance liquid chromatography (HPLC) instrument equipped with a TSK G2500PW column and eluted with deionized water, which filtered through a filter membrane (pore size 0.22 μm) at a flow rate of 0.3 mL/min. A total of 20 μL of 1% sample solutions in deionized water was injected. The molecular weight was evaluated with maltose (MW: 342, 668, 990 Da) and dextran (MW: 2000, 5900, 9600 Da) as standards.

### 4.4. Purification of KSCOs

KSCOs were purified by chromatography using a modified method previously described [80]. The freeze-dried samples were dissolved in 0.02 mol/L NH_4_HCO_3_, and the supernatant after centrifugation (4000 rpm, 10 min) was purified by Bio-Gel P4 chromatography eluting with 0.02 mol/L NH_4_HCO_3_ at a flow rate of 3.0 mL/h. The components collected by the automatic collector were desalted with Bio-GEL P4 column, eluted with 3.0 mL/h distilled water, and freeze-dried after concentration.

### 4.5. Structure Analysis of KSCOs

#### 4.5.1. ESI-MS and TLC Analysis

In order to further analyze the structure, KSCOs were analyzed by ESI-MS in negative ion mode [33]. KSCOs (2.0 mg) were dissolved in acetonitrile: water (1:1, *v*/*v*) to make the concentration within the range of 5–10 pmol/L, and the sample volume was 5 µL. In the process of mass spectrometry, N_2_ was used as the solvent of blow-drying gas and spray gas, and the flow rates were 250 and 15 L/h, respectively. The mobile phase was acetonitrile: water (1:1, *v*/*v*). Driven by the pump, the sample was injected at a flow rate of 10 µL/min. The parameters involved a capillary voltage of 3 kV, a cone-hole voltage of 50 eV, an ionic element volatilization temperature of 80 °C and a solvent volatilization temperature of 150 °C.

The hydrolysate of KSC was analyzed by TLC plate developed with n-butane: ethanol: water (3:2:2, *v*/*v*/*v*) according to the previous description [75]. After drying, the plate was stained with a mixture of vitriol: ethanol (3:17, *v*/*v*; with 0.2% resorcinol, *w*/*v*) and heated until the appearance of clear bands.

#### 4.5.2. Spectroscopy Analysis

FTIR and NMR assays were carried out according to previous methods [80,81]. For FTIR spectra, KSC and its oligosaccharides (2.0 mg) were mixed with KBr (200 mg) powder, ground and pressed, and then measured on a Nicolet Nexus 470 spectrometer (Thermo Fisher Scientific, Waltham, MA, USA). For NMR spectra, KSC (50 mg) was dissolved in 500 μL 99% of the D_2_O, freeze-dried and repeated 3 times. The sample was then redissolved in 500 μL D_2_O and transferred to an NMR tube. Finally, ^1^H-NMR/^13^C-NMR with Agilent DD_2_ 500 MHz NMR spectrometer was performed with acetone as the internal standard.

### 4.6. Induction of UC in Mice and Treatment with KSCOs

#### 4.6.1. Experimental Animals

A total of 40 male C57BL/6 mice (20–22 g) were purchased from Pengyue experimental animal breeding Co., Ltd. (Jinan, China). All animals were raised under the conditions of 20–25 °C, 60–70% relative humidity and 12/12 h light/dark cycle. They were randomly divided into six groups after a one-week acclimatization period (n = 8 per group). All animal experiments were in line with the National Laboratory Animal Ethics Committee of China and were approved by the Animal Care Review Committee (approval number SYXK2020-0422), Qingdao University of Science and Technology, China.

#### 4.6.2. Experimental Procedures

In the normal group, the mice drank water freely from day 0 to day 14. In the DSS group, the mice drank water freely from day 0 to day 7, followed by administration of 3.0% (*w*/*v*) DSS (36 kDa-50 kDa, MP biomedicals) for 7 days. In the KSCOs intervention groups, low-dose (LS, 1.6 mg/kg), medium-dose (MS, 3.2 mg/kg) and high-dose (HS, 6.4 mg/kg) KSCOs were given by gavage every day throughout the experimental cycle and DSS was added to the drinking water from day 7 to day 14. The grouping and respective treatments are detailed in Figure 9. The weight of mice was recorded daily.

#### 4.6.3. Assessment of Colitis

DAI was determined by assessing clinical symptoms including weight loss, fecal traits and hematochezia in mice, then the average of these scores was calculated, as previously described [82]. The specific scoring rules are shown in Table 3. The proximal colon of each group was fixed with 4% paraformaldehyde and embedded in paraffin, which were stained with hematoxylin−eosin (H&E) for histopathological observation.

#### 4.6.4. MPO Activity Analysis

Colon tissues (~0.1 g) were ground in cold normal saline to prepare 10% homogenate. The activity of MPO was measured using homogenate according to the kit (Nanjing Jiancheng, Nanjing, China) instruction.

#### 4.6.5. Level of Cytokines in Serum

The concentrations of interleukin (IL)-6, TNF-α and IL-10 in serum were measured using enzyme-linked immunosorbent assay (ELISA) kits (MultiSciences, Hangzhou, Zhejiang, China) following the manufacturer’s protocol.

#### 4.6.6. SCFAs Analysis

Fecal samples (25 mg) were dissolved in 500 μL of water containing 0.5% phosphoric acid and then were frozen and ground for 3 min (50 HZ), followed by ultrasound for 10 min and centrifugation at 13,000× *g* for 15 min. After that, all of supernatant was removed and n-butanol (0.2 mL) was added to extract SCFAs. Finally, the extract was analyzed by gas chromatograph–mass spectrometer (GC-MS) [59].

#### 4.6.7. Gut Microbiota Analysis

The methods of DNA extraction, PCR amplification and 16S rRNA sequencing were performed as previously described [83]. Genomic DNA was extracted from fecal sample using OMEGA kit and detected by 1% agarose gel electrophoresis. Primers (338F-5′-ACTCCTACGGGAGGCAGCAG-3′ and 806R-5′-GGACTACHVGGGTWTCTAAT-3′) with barcode were synthesized for V3-V4 region amplification of 16S rRNA. Miseq library was constructed and sequenced. PE reads were firstly spliced according to overlap, then the sequence quality was controlled and filtered (Majorbio Bio-Pharm Technology Co. Ltd., Shanghai, China). Operational taxonomic unit (OTU) clustering was performed for nonrepeating sequences according to 97% similarity. Ribosomal database project (RDP) classifier (version 2.13) was used to classify OTU representative sequences. Alpha diversity and Beta diversity were assigned using QIIME software 1.9.1 (Rob Knight, CA, USA). The principal coordinate analysis (PCoA), principal component analysis (PCA) and community structure differences among groups were analyzed with QIIME software and R software 3.5.3 (UoA, AKL, NZ).

### 4.7. Statistical Analysis

The results were expressed as mean ± standard deviation (SD). Data were analyzed via one-way ANOVA with Tukey’s test to determine the statistical significance (*p* < 0.05) using SPSS version 22.0 and GraphPad Prism version 7.0 software (Inc., La Jolla, CA, USA).

## 5. Conclusions

In this study, a κ-selenocarrageenase from the deep-sea bacterium *Bacillus* sp. N1-1 was characterized and expressed in *Escherichia coli*. The reaction temperature was optimized to facilitate the preparation of KSCOs. KSC could be efficiently hydrolyzed by SeCar and yielded a large proportion of small molecular KSCOs (<1500 Da). Spectral analysis showed that selenium oligosaccharides in the hydrolysate of κ-selenocarrageenan were mainly composed of selenium-galactobiose. At present, the application of KSCOs in the treatment of UC is still limited. In this study, the effects of KSCOs administration (1.6 mg/kg, 3.2 mg/kg, 6.4 mg/kg) on UC mice were evaluated. It was suggested that the administration of KSCOs significantly mitigated symptoms of UC, ameliorated neutrophil infiltration and improved inflammatory cytokines dysregulation. We speculated that KSCOs alleviated UC by suppressing inflammatory responses and modulating the composition of gut microbiota. Above all, the κ-selenocarrageenase SeCar could be a potential tool for hydrolyzing κ-selenocarrageenan, and the products of KSCOs were expected to be promising candidates for UC. This study expands the application of organic Se in the treatment of inflammatory diseases.

## Figures and Tables

**Figure 1 ijms-24-04672-f001:**
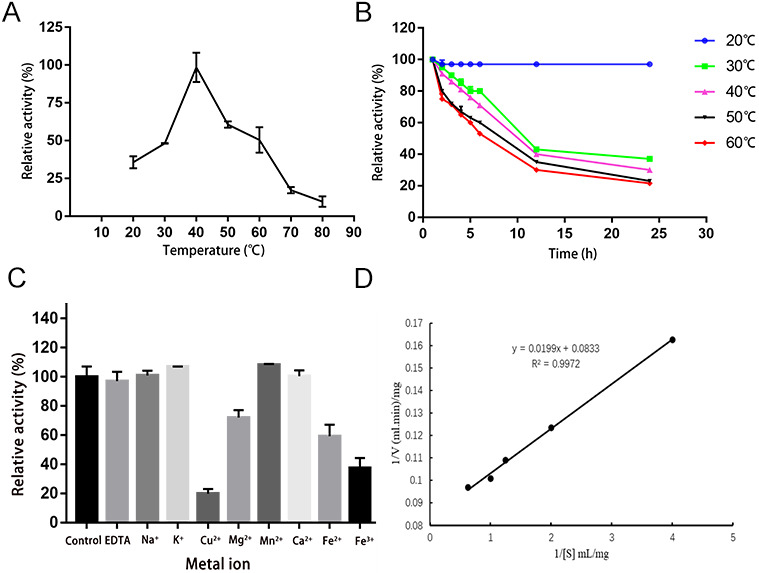
Characterization of SeCar: (**A**) determination of optimum temperature; (**B**) determination of thermotolerance; (**C**) effects of metal ions and chemical reagents on SeCar activity; (**D**) kinetic parameters of SeCar.

**Figure 2 ijms-24-04672-f002:**
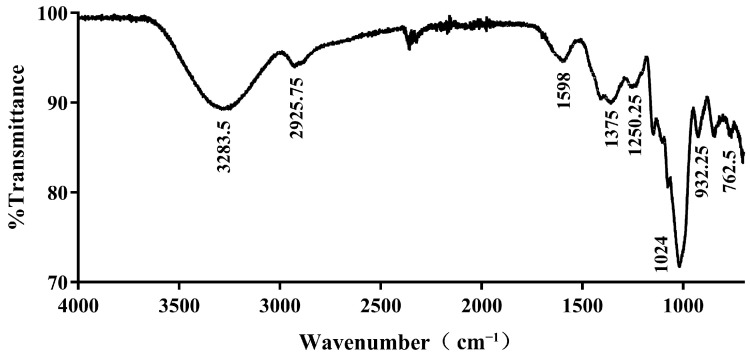
FT-IR spectra of KSCOs.

**Figure 3 ijms-24-04672-f003:**
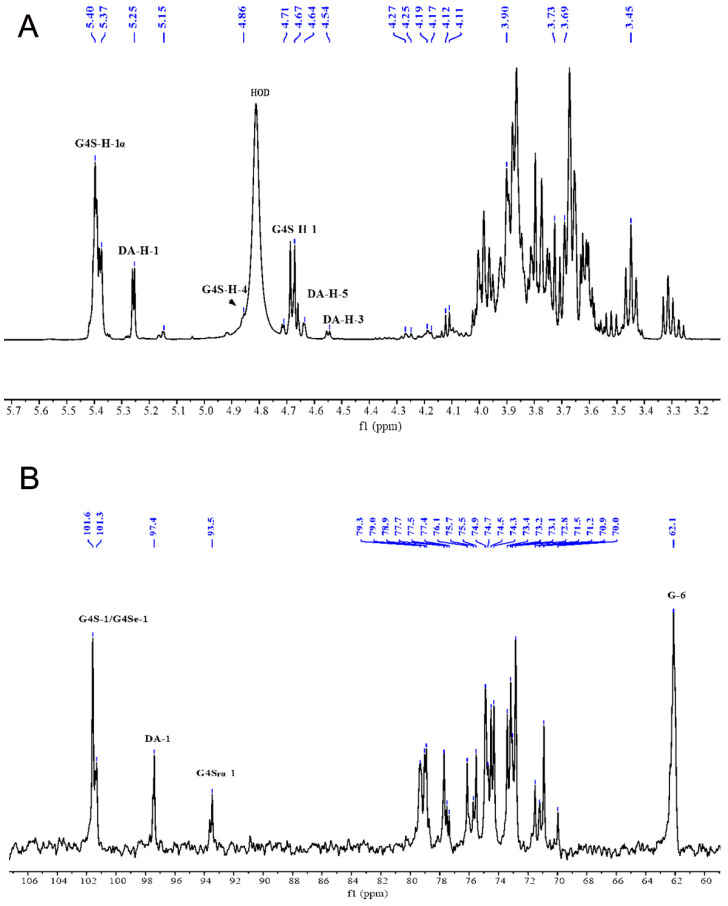
NMR spectra of the KSCOs: (**A**) ^1^H NMR spectrum; (**B**) ^13^C NMR spectra. Blue indicateschemical shift.

**Figure 4 ijms-24-04672-f004:**
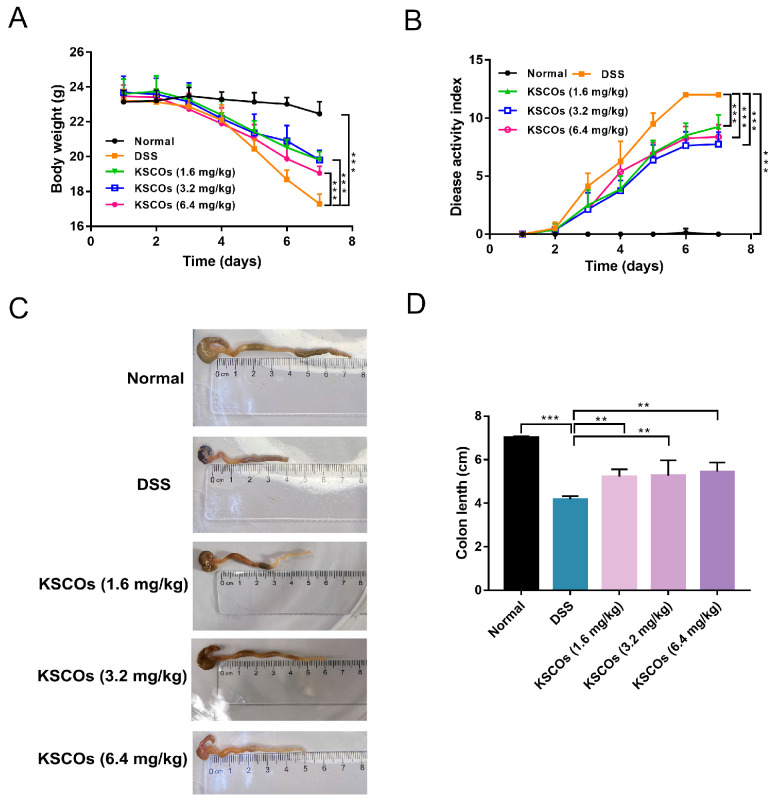
Symptoms of DSS-induced colitis: (**A**) body weight change; (**B**) disease activity index (DAI); (**C**) representative images of colon tissues; (**D**) colon length of different groups. Data are shown as means ± SD, ** *p* < 0.01, *** *p* < 0.001.

**Figure 5 ijms-24-04672-f005:**
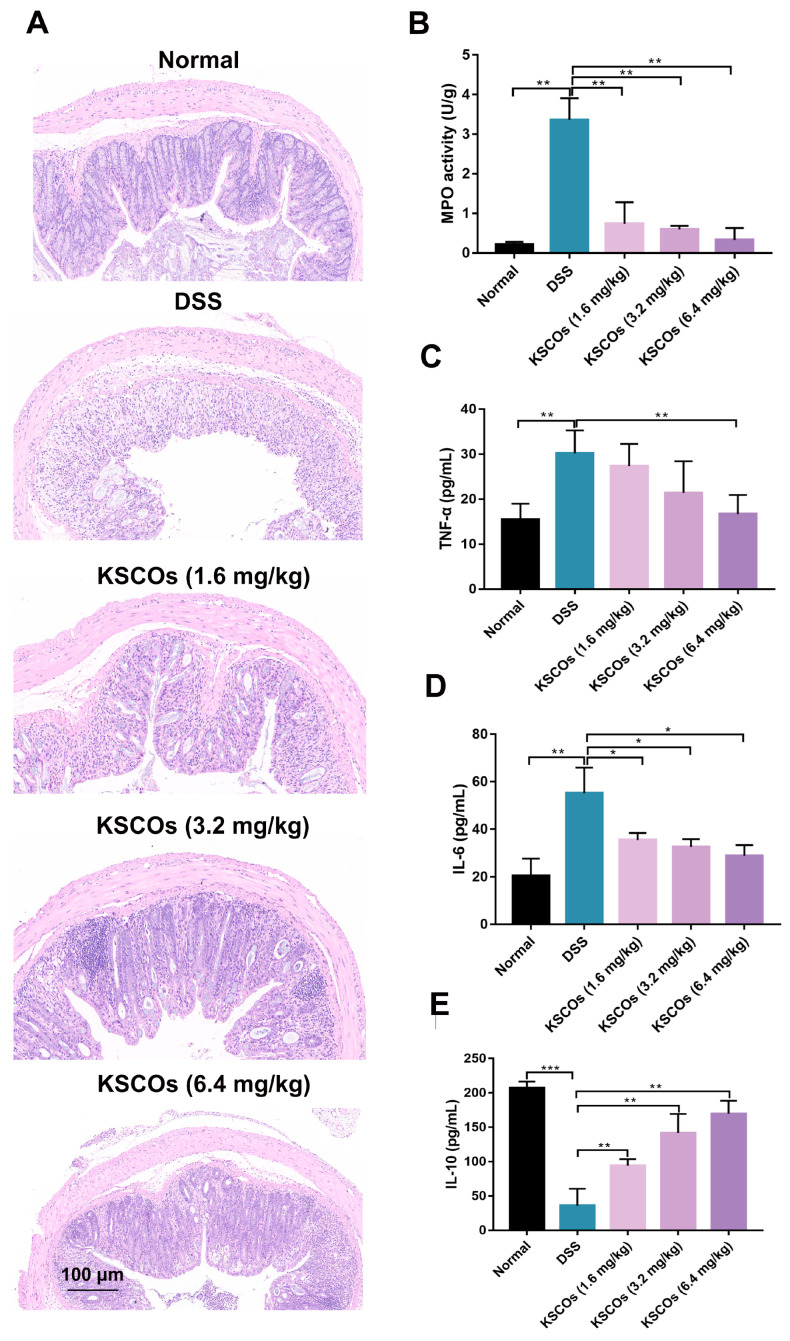
Effect of KSCOs on the histological injury, the activity of MPO in colon tissues and the level of inflammatory cytokines in serum with colitis: (**A**) H&E staining (scale bar = 100 μm); (**B**) MPO activity; (**C**) the contents of TNF-α; (**D**) the contents of IL-6; (**E**) the contents of IL-10. Data are shown as means ± SD, * *p* < 0.05, ** *p* < 0.01, *** *p* < 0.001.

**Figure 6 ijms-24-04672-f006:**
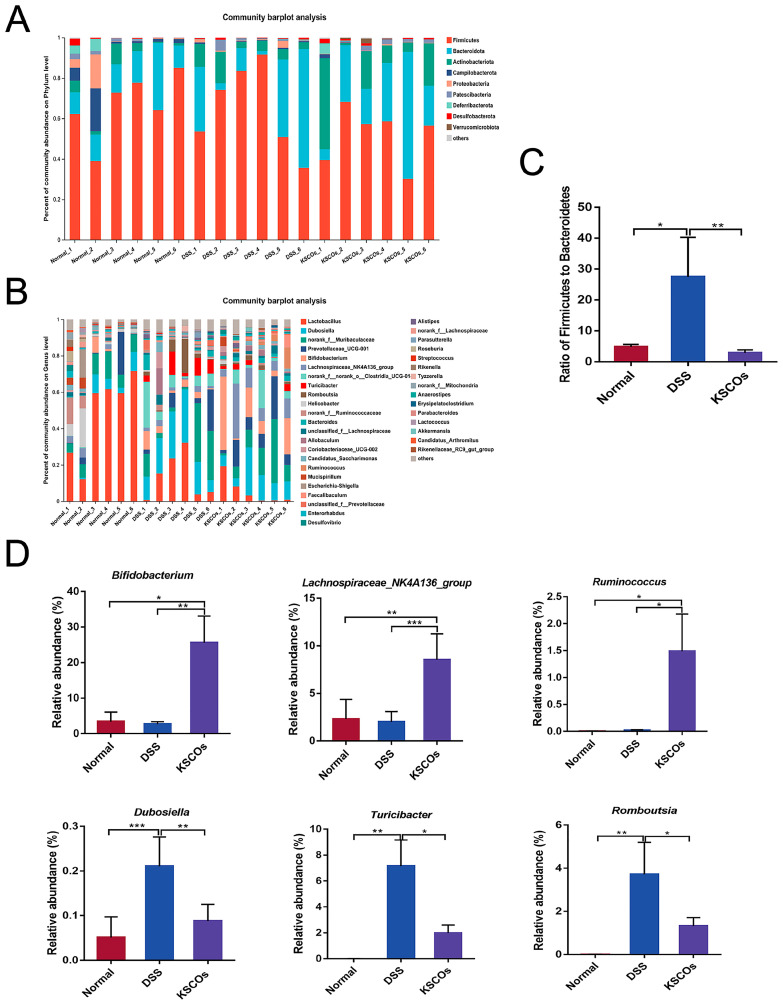
KSCOs modified the structure of gut microbiota: (**A**) the composition of gut microbiota at phylum level; (**B**) the composition of gut microbiota at genus level; (**C**) the ratio of Firmicutes to Bacteroidota (F/B); (**D**) different bacteria at the genus level. Data are shown as means ± SD, * *p* < 0.05, ** *p* < 0.01, *** *p* < 0.001.

**Figure 7 ijms-24-04672-f007:**
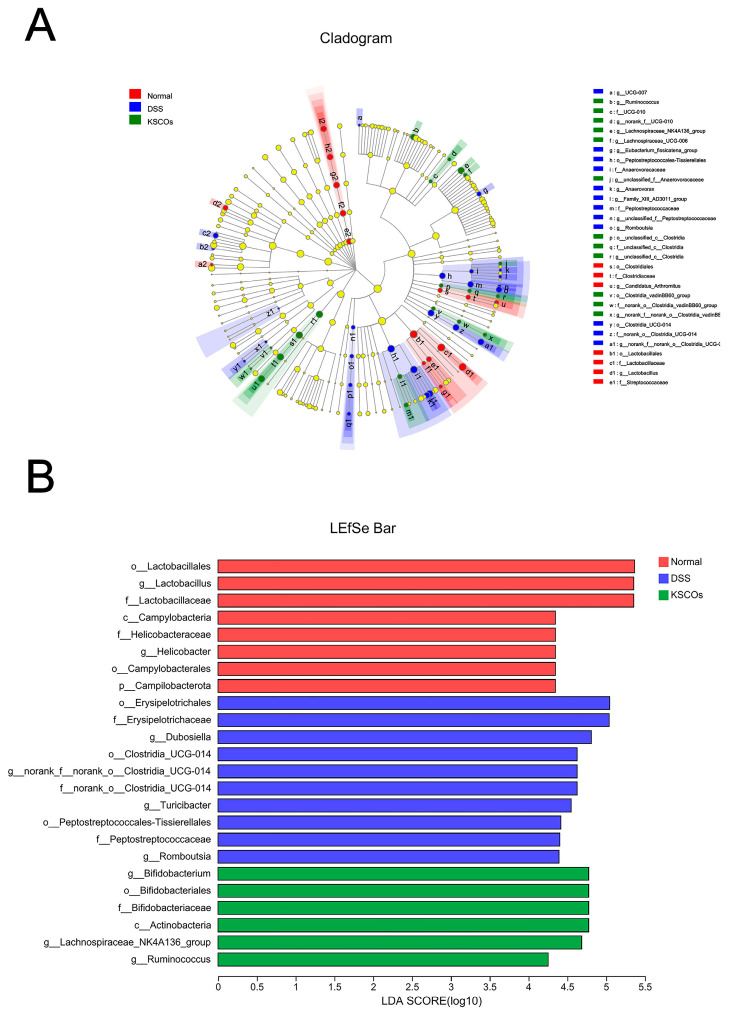
Differences in composition of the gut microbiota among the normal, DSS, and KSCOs groups: (**A**) cladogram of LEfSe analysis; (**B**) histogram of the LDA scores.

**Figure 8 ijms-24-04672-f008:**
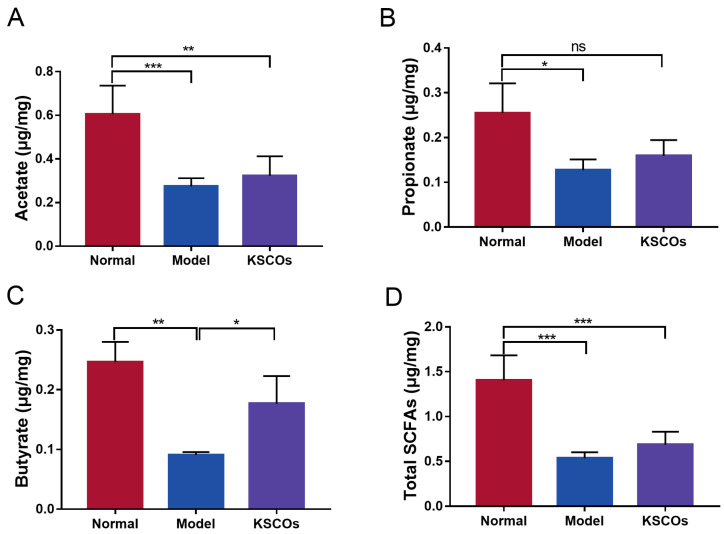
Concentrations of SCFAs in fecal samples: (**A**) the concentration of acetate in each group; (**B**) the concentration of propionate in each group; (**C**) the concentration of butyrate in each group. (**D**) the concentration of total SCFAs in each group. Data are shown as means ± SD, * *p*< 0.05, ** *p* < 0.01, *** *p* < 0.001.

**Figure 9 ijms-24-04672-f009:**
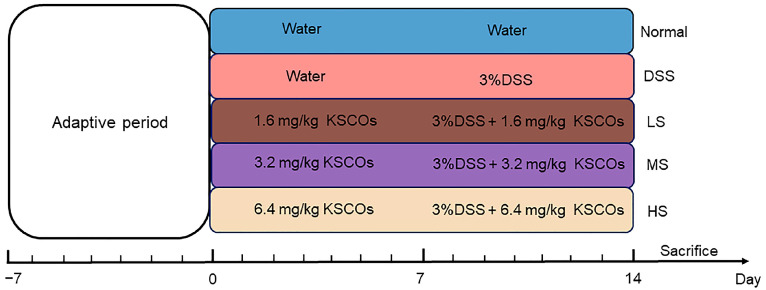
The schematic of experimental procedure.

**Table 1 ijms-24-04672-t001:** ^1^H-NMR chemical shifts of κ-selenocarrageenan oligosaccharides.

Residue	Chemical Shift (ppm)					
	H1	H2	H3	H4	H5	H6
→3)-β-G4S-(1→	4.67	3.45	3.90	4.86	3.73	3.69
→4)-α-DA ^1^-(1→	5.25	4.11	4.55	4.64	4.67	4.17
→3)-G4S ^2^ rα	5.40/5.37					

^1^ 3, 6-anhydrogro-α-1, 4 linked-D-galactose. ^2^ β-1, 3-D-galactose-4-sulfate.

**Table 2 ijms-24-04672-t002:** ^13^C-NMR chemical shifts of κ-selenocarrageenan oligosaccharides.

Residue	Chemical Shift (ppm)					
	C1	C2	C3	C4	C5	C6
→3)-β-G4S-(1→	101.6	71.5	79.3	73.2	75.5	62.1
→3)-β-G4Se-(1→	101.3	71.2	79.0	73.1	74.9	62.1
→4)-ɑ-DA-(1→	97.4	70.9	79.0	78.9	75.7	70.0
→3)-G4Srα	93.5	70.0	76.1	73.4	74.3	62.1

**Table 3 ijms-24-04672-t003:** The specific scoring rules of the disease activity index (DAI).

Parameters		Score
Weight loss	0%	0
	1–5%	1
	6–10%	2
	11–15%	3
	>15%	4
Fecal traits	Normal	0
	Soft stools	2
	Diarrhea	4
Hematochezia	Normal	0
	Presence of blood	2
	Abundant bleeding	4

## Data Availability

Not applicable.

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
