# Peer review of "κ-Selenocarrageenan Oligosaccharides Prepared by Deep-Sea Enzyme Alleviate Inflammatory Responses and Modulate Gut Microbiota in Ulcerative Colitis Mice"

_ijms, 2023, doi:10.3390/ijms24054672_

Round 1
Reviewer 1 Report
The manuscript titled, κ-Selenocarrageenan oligosaccharides prepared by deep-sea enzyme alleviate inflammatory responses and modulate gut microbiota in ulcerative colitis mice is very well written. However, methodologies must be with appropriate citation and other details. Figures quality must be improved
Abstract is clear with appropriate findings
Introduction
This section needs to be improved by highlighting the need of conducting this study. Research objectives are not clear, which should be clearly stated at the end of the section
Materials and Methods
4.1.2. Expression and purification… provide citation
Some methodologies need further information or provide citations. Each methodology should provide enough information to repeat the experiments.
4.2. Isolation of the KSCOs… provide citation
Purification of KSCOs : provide citation
4.5. Structure analysis of KSCOs: provide citation
There are many methodologies without citations. This is not the way to write scientific manuscript
4.6.7. Gut microbiota analysis: provide citation
Figure 9 : low quality and not readable
Results
Although the results are interesting, almost all figures are low quality and it is difficult to evaluate the results based on figures. Authors must improve the quality of all figures.
Discussion is appropriate and compared with literature
References are not according to the journal format. Please revise it
Reviewer 2 Report
κ-Carrageenan selenide (KSC) is a kind of polysaccharide which is difficult to hydrolyze, so it has certain potential in the treatment of intestinal diseases. κ-selenated carrageenan oligosaccharides (KSCOs) were prepared by enzymatic hydrolysis of KSC by SeCar.
In this manuscript, the authors summarized and discussed the effects of KSCOs on dextran sulfate sodium (DSS)-induced ulcerative colitis (UC) in C57BL/6 mice. On this basis, the author conducted research from the aspects of testing inflammatory factors, relieving the symptoms of enteritis and regulating the composition of intestinal flora. KSCOs were expected to be a promising candidate for UC. The anti-inflammation activity study of KSCOs provides the foundational information and applications of organic Se in the treatment diseases.
However, before accepting, several errors and confusing questions need to be paid attention to. For details, see the following comments.
1. The manuscript is not readable and it is recommended that it be read by a native English speaker or polished by a commercial company.
2. The figure labels in Fig. 3, 6, and 7 are not clearly visible.
3. Adding western blotting to the experiment would be more convincing in the anti-inflammatory part.
4. In line 362, “7500´g”should be “7500 ´ g”.
Reviewer 3 Report
In this study, a deep-sea bacteria was used to produce an enzyme called κ-selenocarrageenase (SeCar), which was able to degrade the organic selenium polysaccharide κ-Selenocarrageenan (KSC). The degradation produced κ-selenocarrageenan oligosaccharides (KSCOs) composed mainly of selenium-galactobiose. The study then tested the effects of KSCOs on dextran sulfate sodium (DSS)-induced ulcerative colitis (UC) in mice and found that KSCOs could alleviate the symptoms of UC, reduce the activity of myeloperoxidase, and regulate the gut microbiota to prevent or treat UC. In summary, the authors conducted extensive experiments and this reviewer sees no reason for non-publication.
Author Response
We sincerely thank you for your enthusiastic work and for approving our paper.